# Genotype–Phenotype Analysis of Children with Epilepsy Referred for Whole-Exome Sequencing at a Tertiary Care University Hospital

**DOI:** 10.3390/children10081334

**Published:** 2023-08-01

**Authors:** Fahad A. Bashiri, Rawan AlSheikh, Muddathir H. Hamad, Hamad Alsheikh, Rana Abdullah Alsheikh, Amal Kentab, Najd AlTheeb, Malak Alghamdi

**Affiliations:** 1Department of Pediatrics, College of Medicine, King Saud University, Riyadh 11461, Saudi Arabia; akentab@ksu.edu.sa (A.K.); malghamdi@ksu.edu.sa (M.A.); 2Division of Pediatric Neurology, Department of Pediatrics, King Saud University Medical City, Riyadh 11461, Saudi Arabia; mudhamad@ksu.edu.sa (M.H.H.); 7amad.alsheikh@gmail.com (H.A.); 3Division of Pediatric Neurology, Department of Pediatrics, King Saud Medical City, Riyadh 11461, Saudi Arabia; r.alsheikh@ksmc.med.sa (R.A.); dralsheikh@ksmc.med.sa (R.A.A.); 4College of Medicine, King Saud University, Riyadh 11461, Saudi Arabia; 5Division of Medical Genetics, Department of Pediatrics, King Saud University Medical City, Riyadh 11461, Saudi Arabia

**Keywords:** epilepsy, children, genetics, mutations, genotype, phenotype, Saudi Arabia

## Abstract

Background: Despite the high consanguinity rates, data on genetic epilepsy in Saudi Arabia is limited. The objective of the current study was to characterize genetic mutations associated with epilepsy in pediatric patients and describe their phenotypic presentations. Methods: A retrospective chart review was conducted among children presented with epilepsy in one center in Saudi Arabia between 2015 and 2018. Only those who had undergone genetic testing were included. Results: A total of 45 patients had positive whole-exome sequencing (WES) genetic testing with 37 mutations. Six mutations (SCN1A, DENND5A, KCNQ2, ACY1, SCN2A, and PCDH19) were repeated in 15 patients, with largely heterogeneous phenotypic presentations in patients with the same mutation. Several mutations are reported for the first time in Saudi Arabia. The median age at epilepsy onset was four months. Consanguineous parents and family history of epilepsy were frequent (31.8% and 33.3%, respectively). Developmental delay (44.4%), cognitive delay (42.2%), language delay (40.0%), behavioral features (28.9%), and microcephaly (20.0%) were frequent presentations. At initial diagnosis, 68.9% of EEG and 48.9% of brain MRI were abnormal. The most currently used antiseizure medications (ASMs) were levetiracetam (48.9%), topiramate (28.9%), and valproic acid (20.0%). Approximately 60% of the patients were controlled with (47.6%) or without (11.9%) ASMs, and three (7.1%) patients died. Conclusions: Multiple mutations among children with epilepsy are reported in one hospital in Saudi Arabia, with the majority reported for the first time. The current findings highlight the importance of doing genetic testing for the evaluation of childhood epilepsy.

## 1. Introduction

Epilepsy is defined as a disorder of the brain characterized by an enduring predisposition to generate epileptic seizures and by the neurobiological, cognitive, psychological, and social consequences of this condition manifested with at least one epileptic seizure [1]. The incidence of epilepsy in children worldwide is estimated to be between 35 and 187 new cases per 100,000 and is consistently highest in the first year of life [2,3]. The prevalence of epilepsy in children ranges between 3 and 7 active cases per 1000, with much higher rates in developing countries [2,3]. In Saudi Arabia, the prevalence was estimated to be 6.5 per 1000 in children under ten years, with much higher rates in males than females [4]. With long-term medications, frequent outpatient and emergency visits, and hospitalization, epilepsy constitutes a considerable financial burden for family and society [5,6]. Furthermore, it increases the risk of death among children, particularly those with other neurodevelopmental disorders [7,8].

According to current investigations, approximately half of the epilepsies are of unknown causes [9]. Genetic causes were identified in 22% of epilepsies and are expected to explain an additional portion of the unknown cases with advanced technologies [9,10,11]. The genetic etiology of epilepsy may be based solely on a family history of an autosomal dominant disorder or identification of the causative mutation [12]. Numerous genetic mutations have been identified to cause epilepsy [13]. Most genes show phenotypic heterogeneity, with a single gene mutation causing a spectrum of mild to severe epilepsies [14,15].

Additionally, most epilepsy syndromes reveal genetic heterogeneity with the same syndrome that can be caused by one or more of several genes [11,13,16]. Detecting genetic mutations and understanding the underlying biological mechanism can improve the prognosis by selecting the appropriate ASMs [17,18]. Genetic testing has been suggested for assessing and managing pediatric drug-resistant epilepsy [16,19]. Genetic testing is essential for counseling and predicting the outcome and the risk of recurrence [16,19].

In Saudi Arabia, at least 13% of early childhood epilepsy has genetic etiology [20]. Higher consanguinity rates may indicate a higher contribution of abnormal genes to childhood epilepsy in Saudi Arabia [21]. Nashabat et al. described a series of 72 patients presented with early infantile epileptic encephalopathy with variable presentations ranging from benign to severe courses [22]. Despite some attempts to describe case reports and familial clusters of genetic epilepsy [23,24,25,26,27,28,29], describing the genotype–phenotype correlation in children with genetic epilepsy is still lacking. The presence of comorbidities like autism spectrum disorders or specific learning disorders will increase the possibility of determining underlying genetic etiology in children with epilepsy [30]. The objective of the current study was to characterize genetic mutations associated with epilepsy in pediatric patients and describe the presentation, management, and outcome of patients with confirmed genetic mutations.

## 2. Materials and Methods

The current study was conducted at King Khalid University Hospital in Riyadh (KKUH), Saudi Arabia, pediatric neurology, and genetic clinics. It was a retrospective chart review cross-sectional study conducted in 2019. The study design obtained the ethical approvals of the ethical review committee board at King Saud University, College of Medicine, Riyadh, Saudi Arabia. 

All pediatric patients aged ≤ 18 years who presented with epilepsy in pediatric neurology and genetic clinics of KKUH between May 2015 and December 2018 and were subjected to genetic testing using whole-exome sequencing (WES) were included in the study.

Demographic data, age of onset of seizures, seizure types at initial diagnosis, baseline development before seizure onset, family history of epilepsy, and consanguinity were considered. Other associated neurological features are also described. electroencephalogram (EEG)and Brain MRI reports were described in both WES-positive and WES-negative children. Antiseizure medication and seizure outcomes are also described. The genetic testing result was reviewed, and the pathogenicity of variants was classified using the American College of Medical Genetics and Genomics (ACMG). 

### 2.1. Data Collection Tool

A structured questionnaire was used in data collection. It included questions about age, gender, nationality, parents’ consanguinity, family history, age at initial diagnosis, type of seizures, neurological features, cognitive/language delay, abnormal behavioral features, electroencephalogram (EEG) reports, brain magnetic resonance imaging (MRI) reports, ASMs used, seizure outcomes, and results of genetic testing. 

### 2.2. Statistical Analysis

Categorical variables are presented as frequencies and percentages. Continuous variables are presented as means and standard deviations (SDs) when normally distributed and median and inter-quartile range (IQR) when not normally distributed. The differences in age, gender, presentations, EEG and MRI reports, treatment, and outcomes were compared between those with and without mutations. The chi-squared test or Fisher’s exact test, as appropriate, was used to evaluate categorical differences, and the *t*-test or Mann–Whitney test, as appropriate, was used to evaluate continuous differences. All *p*-values were two-tailed. A *p*-value < 0.05 was considered significant. Statistical Package for the Social Sciences software (SPSS Version 25.0. Armonk, NY, USA: IBM Corp) was used for all statistical analyses.

## 3. Results

Of 294 children with epilepsy reviewed, 53 were referred for genetic analysis. Out of the 53 results, 45 (84.9%) had one or more related genetic mutations, including pathogenic, likely pathogenic, or variant of uncertain significance, with strong evidence, and 8 (15.1%) had negative WES findings.

The current description focused on those with positive WES compared to negative WES. The average age of patients with mutations was 9.4 ± 7.1 years, 51.1% were females, and 97.8% had Saudi nationality. The patients were equally presented at initial diagnosis with focal and generalized epilepsy (31% and 29%, respectively) and infantile spasm (15.6%). The generalized epilepsy was mainly tonic–clonic (9/13). Later presentations were mostly generalized epilepsy (56%) and, to a lesser extent, focal epilepsy (24%). The median age at initial diagnosis was four months (interquartile range 1.4–13.5 months), and 75.5% had the onset in the first year. Psychomotor delay before the initial diagnosis of epilepsy was found in 70.0% of the patients. Almost a third of the patients had consanguineous parents (31.8%) and a family history of epilepsy (33.3%). Compared with negative WES, those with mutations had more consanguineous parents, younger age at onset of epilepsy, and more generalized and focal epilepsy at initial diagnosis. However, none of these differences were statistically different (*p* = 0.412, *p* = 0.286, and *p* = 0.120, respectively) (Table 1 shows demographic characteristics and medical history).

Three-quarters (75.6%) of the patients with mutations had one or more neurological features. These included developmental delay (44.4%), microcephaly (20.0%), hypotonia (15.6%), dysmorphic features (15.6%), vision problems (11.1%), and movement disorders (11.1%). Almost two-thirds had a cognitive delay (24.4%), language delay (22.2%), or both cognitive/language delay (17.8%). Approximately 29% of the patients had one or more abnormal behavioral features. These included attention deficit hyperactivity disorder (ADHD, 8.9%) and autism spectrum disorder (ASD, 4.4%). Compared with those with negative WES, those with mutations had more developmental delay, dysmorphic features, and abnormal behavioral features but less ataxia and choreoathetosis. Except for ataxia (*p* = 0.020), none of these differences reached statistical significance (Table 2 shows the patients’ Neurological, cognitive, and behavioral features). The patient with FARS2 gene mutation had intractable focal onset seizure since birth, hypotonia, and developmental delay. His MRI is shown in Figure 1. On the other hand, the ADAT3 patient had his first seizure at six years of age in form of generalized tonic–clonic seizure, which evolved later into staring and behavior arrest. His seizure was later on controlled with no medication. He had global developmental delay before seizure onset. His brain MRI was unremarkable. 

Abnormal EEG findings among patients with mutations were 68.9% at initial diagnosis and 46.3% at follow-up. Epileptiform discharges were more commonly seen than abnormal background activities at initial diagnosis but were equally seen at follow-up. The patient with SCN2A had slow background activity with a burst-suppression pattern (Figure 2). Almost half (48.9%) of the patients had abnormal brain MRI, which included generalized brain atrophy (45.5%), abnormal signal intensity (27.3%), thinning/dysgenesis of the corpus callosum (27.3%), and ventriculomegaly (22.7%). These findings were more common (not statistically significant) in patients with mutations than those with negative WES.

Table 3 summarizes electroencephalogram (EEG) and magnetic resonance imaging (MRI) findings.

The most common ASMs currently used for the patients with mutations were levetiracetam (48.9%), topiramate (28.9%), valproic acid (20.0%), and phenobarbitone (17.8%). About 60% of the patients were controlled with (47.6%) or without (11.9%) ASMs. Death was reported in (7.1%). Table 4 summarizes Antiseizure medications (ASMs) used and the outcome.

Out of 45 patients, 37 had positive WES. Six different mutations (SCN1A, DENND5A, KCNQ2, ACY1, SCN2A, and PCDH19) were repeated in 15 patients, while the rest of the mutations were non-repeated mutations in 31 patients. All the six genes were reported previously with developmental and epileptic encephalopathy or neurometabolic condition. (The genotype–phenotype analysis of the patients with frequent mutations is detailed in Table 5).

## 4. Discussion

We report genetic mutations and associated phenotypes among pediatric patients with epilepsy seen over three years in one center in Saudi Arabia. Four mutations seen in multiple patients in the current cohort have been reported before in Saudi Arabia. sodium channel, neuronal type 1, alpha subunit (SCN1A) mutation, known to be associated with a broad spectrum of infant/childhood epilepsy [14], was detected in four patients in the current study. Three of them were from the same family and had mild presentation, while the fourth had severe “Dravet syndrome” starting at two months of age. SCN1A mutation was previously reported in a Saudi girl diagnosed with Dravet syndrome who had seizures at eight months of age in response to fever, illness, and diaper changing [23].

Similarly, sodium channel, neuronal type 2, alpha subunit (SCN2A) mutation, which is known to be associated with a broad spectrum of infant/childhood epilepsy [15], was detected in two patients in the current study, with mild presentation that started in the first three months of life. SCN2A mutation has been reported in two Saudi girls who presented with early infantile epileptic encephalopathy starting in the first few days of life [24,31]. Additionally, DENND5A mutation was reported in the current study in three patients from the same family who presented with Niemann–Pick syndrome. The same gene has been reported in 2 out of 337 in the Saudi cohort with intellectual disability [29] and 2 Saudi sisters with epileptic encephalopathy [32]. Finally, KCNQ2 (potassium voltage-gated channel subfamily Q member 2) mutation was detected in two patients in the current study who presented with focal seizures between the first and fourth months. KCNQ2 has been reported in three Saudi patients with epileptic seizures starting in the first four months of life [26]. The phenotypes of most of the described cases are consistent with the genotype. There are no specific criteria to indicate genetic testing or the underlying genetic etiology. However, most cases of epilepsy with underlying genetic etiology presented early in life, associated with additional features like microcephaly, global developmental delay, facial dysmorphism, and vision problems, and have abnormal MRI features. Usually, the family history is suggestive, like consanguineous marriage and previously affected siblings or cousins. In some cases, genetic diagnosis modifies the treatment plan, like in a female who presented with neonatal epileptic encephalopathy, and we identified a de novo variant in the SCN2A gene which resulted in modifying the therapy to a ketogenic diet with good response. Most of the identified variants were missense variants and identified in de novo status, then recessive genes and few as X-linked, like the PCDH19 and PLP1 genes.

Three genetic mutations detected in single patients in the current study were reported before as familial clusters or a cohort of single-gene mutation in Saudi Arabia. In the current study, potassium inwardly rectifying channel subfamily J member 10 (KCNJ10) mutation was detected in one patient who presented with drop attacks, movement disorder, and intellectual disability. KCNJ10 mutation was also reported in five Saudi patients from two families who presented with EAST syndrome, epilepsy, ataxia, and sensorineural deafness [25]. Additionally, seizure threshold 2 (SZT2) mutation was detected in one patient in the current study who presented with generalized tonic–clonic seizures at four months of age without developmental delay. SZT2 was also reported in two Saudi sisters with epilepsy, developmental delay, and macrocephaly [27]. Finally, tuberous sclerosis complex 2 (TSC2) mutation was detected in one patient in the current study who presented with focal seizures at birth and later global developmental delay. TSC2 was detected in 65% of a cohort of patients with genetically confirmed tuberous sclerosis complex, including seizures, skin manifestations, and developmental delay [28]. 

Several currently detected mutations are reported for the first time among Saudi patients. For example, ACY1, known to cause a rare inborn error of metabolism [33], was reported in two patients from the same family who presented with infantile spasms and severe developmental and intellectual delay. Furthermore, PCDH19, known to cause epilepsy with mental retardation [34], was reported in two unrelated patients with developmental delay and intellectual disability. Finally, two mitochondrial mutations affecting transfer RNAs (FARS2 and ADAT3) detected in the current study have never been reported before among Saudi patients. 

Comparing the clinical presentation and management of the patients with confirmed genetic mutations in the current study with local studies is difficult as the majority of previous reports were either case reports or clusters of single-gene mutation [23,24,25,26,27], with very limited data on genetic epilepsy of different mutations [20]. Additionally, phenotypic heterogeneity of other genes further complicates any comparison [12,13]. Developmental/cognitive delay, mainly starting before the onset of seizures, was frequent and probably more common than in previous studies [20,35]. As expected, microcephaly, hypotonia, and dysmorphic features were the common presentations, and epilepsy started within the first year [13,35,36]. Similar to previous studies [21], family history was common, and most of the repeated genetic mutations in the current study were among family members. Although parents’ consanguinity in the present study was more common than in Western studies, it was less prevalent than in other Saudi studies examining the prevalence of epilepsy [37]. The death rate in the current patients may be higher than reported before [7,8], but the interpretation is difficult without adjustment for other comorbidities.

In patients with drug-resistant epilepsy, genetic testing is crucial to target the choices of antiseizure medication (ASM) and predict the response to ASM. Margari et al. studied the Association between SCN1A gene polymorphisms and drug-resistant epilepsy in pediatric patients and showed that the intronic rs6730344, rs6732655, and rs10167228 polymorphisms of the SCN1A gene are potential risk factors for drug resistance. Additionally, AA and AT genotypes of the rs1962842 intronic polymorphism also emerged as a risk factor in the drug-resistant group. Therefore, polymorphisms of the SCN1A gene could play a role in the response to AED in patients with drug-resistant epilepsy, with important implications for clinical practice [38].

The current study is considered the first local study to examine genetic epilepsy caused by multiple genetic mutations among children. Additionally, it assessed the genotype–phenotype correlations and reported several mutations for the first time in Saudi Arabia. Nevertheless, many limitations are acknowledged. Although it included all patients with genetic epilepsy in one center over three years, the sample size was relatively small. The retrospective design used cannot confirm causality. Although these limitations are unlikely to have affected the findings, they may point to the need for a future multicenter prospective study on genetic epilepsy among children in Saudi Arabia.

In conclusion, multiple mutations among children with epilepsy are reported in one hospital in Saudi Arabia, with the majority reported for the first time. Heterogeneous phenotypic presentations were frequently seen in epileptic patients with the same mutation. The majority had early-onset epilepsy. The most frequent presentations were developmental, cognitive/language delay, behavioral features, and microcephaly. A multicenter prospective study may be justified to understand different genotype–phenotype presentations of genetic epilepsy.

## Figures and Tables

**Figure 1 children-10-01334-f001:**
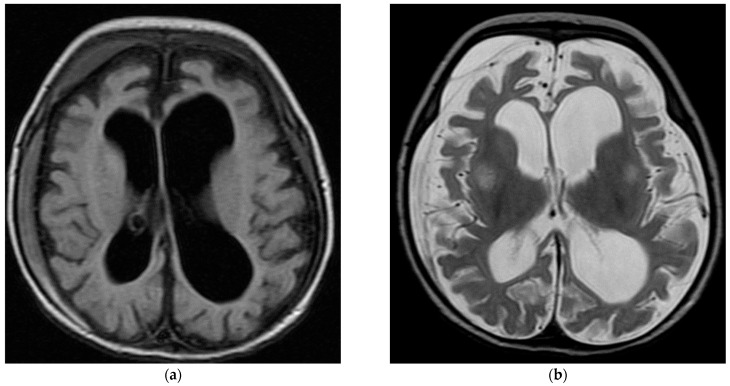
Brain MRI for patient with FARS2 gene. (**a**) AX FLAIR images showing global cerebral atrophy, small bilateral subdural hematoma more on the right side. (**b**) AX T2 showing near symmetrical bilateral abnormal high T2 signal intensity involving the basal ganglia more pronounced at the putamen.

**Figure 2 children-10-01334-f002:**
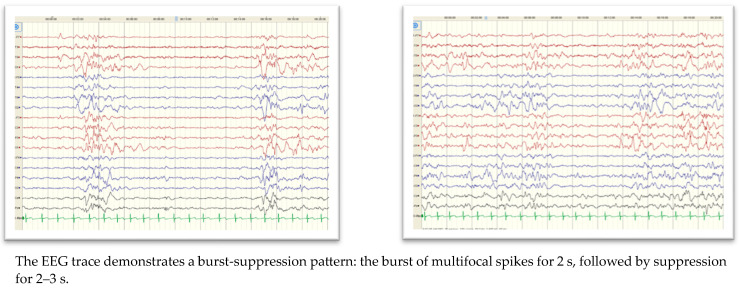
EEG for patient with SCN2A.

**Table 1 children-10-01334-t001:** Demographic characteristics and medical history of pediatric patients with genetic testing (N = 53).

	Mutation (N = 45)	No Mutation (N = 8)	Total (N = 53)	*p*-Value
**Age (years)**				
Mean ± standard deviation	9.4 ± 7.1	9.3 ± 7.3	9.3 ± 7.2	0.952
Median and inter-quartile range	7 (4–15)	9 (3.5–12.5)	7 (4–15)	0.893
**Gender**				
Male	22 (48.9%)	4 (50.0%)	26 (49.1%)	>0.99
Female	23 (51.1%)	4 (50.0%)	27 (50.9%)	
**Seizure type at initial diagnosis**				
Generalized, tonic–clonic	9 (20.5%)	1 (12.5%)	10 (19.2%)	0.120
Generalized, absence	3 (6.8%)	0 (0.0%)	3 (5.8%)	
Generalized, non-specified	1 (2.3%)	0 (0.0%)	1 (1.9%)	
Focal	14 (31.8%)	1 (12.5%)	15 (28.8%)	
Infantile spasm	7 (15.9%)	1 (12.5%)	8 (15.4%)	
Drop attack	2 (4.5%)	0 (0.0%)	2 (3.8%)	
Nocturnal seizure	0 (0.0%)	2 (25.0%)	2 (3.8%)	
Others	8 (18.2%)	3 (37.5%)	11 (21.2%)	
**Other seizure type**				
Generalized, tonic–clonic	12 (48.0%)	0 (0.0%)	12 (44.4%)	0.140
Generalized, absence	1 (4.0%)	0 (0.0%)	1 (3.7%)	
Generalized, non-specified	1 (4.0%)	0 (0.0%)	1 (3.7%)	
Focal	6 (24.0%)	1 (50.0%)	7 (25.9%)	
Infantile spasm	0 (0.0%)	1 (50.0%)	1 (3.7%)	
Drop attack	2 (8.0%)	0 (0.0%)	2 (7.4%)	
Nocturnal seizure	0 (0.0%)	0 (0.0%)	0 (0.0%)	
Others	3 (12.0%)	0 (0.0%)	3 (11.1%)	
**Age at onset of epilepsy (months)**				
Median and inter-quartile range	4 (1.4–13.5)	6 (6–17.2)	5 (2.0–13.5)	0.286
**Development before seizure onset**				
Normal	12 (30.0%)	3 (37.5%)	15 (31.3%)	0.692
Delayed	28 (70.0%)	5 (62.5%)	33 (68.8%)	
**History**				
Parents consanguinity	14 (31.8%)	1 (12.5%)	15 (28.8%)	0.412
Family history of epilepsy	15 (33.3%)	2 (25.0%)	17 (32.1%)	>0.99

**Table 2 children-10-01334-t002:** Neurological, cognitive, and behavioral features of pediatric patients with genetic testing (N = 53).

	Mutation (N = 45)	No Mutation (N = 8)	Total (N = 53)	*p*-Value
**Neurological features**				
None	11 (24.4%)	3 (37.5%)	14 (26.4%)	0.882
1–2 features	22 (48.9%)	3 (37.5%)	25 (47.2%)	
3 or more features	12 (26.7%)	2 (25.0%)	14 (26.4%)	
**Neurological features**				
Developmental delay	20 (44.4%)	2 (25.0%)	22 (41.5%)	0.445
Microcephaly	9 (20.0%)	1 (12.5%)	10 (18.9%)	>0.99
Hypotonia	7 (15.6%)	1 (12.5%)	8 (15.1%)	>0.99
Dysmorphic features	7 (15.6%)	0 (0.0%)	7 (13.2%)	0.577
Movement disorders	5 (11.1%)	1 (12.5%)	6 (11.3%)	>0.99
Vision problems	5 (11.1%)	0 (0.0%)	5 (9.4%)	>0.99
Choreoathetosis	2 (4.4%)	2 (25.0%)	4 (7.5%)	0.104
Nystagmus	3 (6.7%)	1 (12.5%)	4 (7.5%)	0.491
Speech problems	3 (6.7%)	1 (12.5%)	4 (7.5%)	0.491
Ataxia	0 (0.0%)	2 (25.0%)	2 (3.8%)	0.02
Spastic quadriplegia	2 (4.4%)	0 (0.0%)	2 (3.8%)	>0.99
Others	13 (28.9%)	1 (12.5%)	14 (26.4%)	0.665
**Cognitive/language delay**				
Cognitive	11 (24.4%)	2 (25.0%)	13 (24.5%)	0.953
Language	10 (22.2%)	2 (25.0%)	12 (22.6%)	
Cognitive/language	8 (17.8%)	2 (25.0%)	10 (18.9%)	
Neither	16 (35.6%)	2 (25.0%)	18 (34.0%)	
**Abnormal behavioral features**				
Any feature	13 (28.9%)	1 (12.5%)	14 (26.4%)	0.665
Attention deficit hyperactivity disorder (ADHD)	4 (8.9%)	0 (0.0%)	4 (7.5%)	>0.99
Autism spectrum disorder (ASD)	2 (4.4%)	0 (0.0%)	2 (3.8%)	>0.99
Hyperactivity disorder	2 (4.4%)	0 (0.0%)	2 (3.8%)	>0.99
Aggressive behavior	1 (2.2%)	0 (0.0%)	1 (1.9%)	>0.99
Anxiety disorder	1 (2.2%)	0 (0.0%)	1 (1.9%)	>0.99
Phobia	1 (2.2%)	0 (0.0%)	1 (1.9%)	>0.99
Other abnormal behavior	3 (6.7%)	1 (12.5%)	4 (7.5%)	0.491

**Table 3 children-10-01334-t003:** Findings of electroencephalogram (EEG) and brain magnetic resonance imaging (MRI) conducted among pediatric patients with genetic testing (N = 53).

	Mutation (N = 45)	No Mutation (N = 8)	Total (N = 53)	*p*-Value
**EEG at initial diagnosis**				
Normal	10 (22.2%)	1 (12.5%)	11 (20.8%)	0.490
Abnormal	31 (68.9%)	5 (62.5%)	36 (67.9%)	
Not available	4 (8.9%)	2 (25.0%)	6 (11.3%)	
**EEG at initial diagnosis, positive findings**				
Abnormal background activity	5 (16.1%)	1 (20.0%)	6 (16.7%)	0.450
Epileptiform discharges	11 (35.5%)	3 (60.0%)	14 (38.9%)	
Both	15 (48.4%)	1 (20.0%)	16 (44.4%)	
**Follow-up EEG**				
Normal	13 (31.7%)	1 (12.5%)	14 (28.6%)	0.473
Abnormal	19 (46.3%)	4 (50.0%)	23 (46.9%)	
No follow-up	9 (22.0%)	3 (37.5%)	12 (24.5%)	
**Follow-up EEG, positive findings**				
Abnormal background activity	6 (31.6%)	0 (0.0%)	6 (26.1%)	0.419
Epileptiform discharges	6 (31.6%)	1 (25.0%)	7 (30.4%)	
Both	7 (36.8%)	3 (75.0%)	10 (43.5%)	
**Brain MRI**				
Normal	22 (48.9%)	2 (25.0%)	24 (45.3%)	0.373
Abnormal	22 (48.9%)	6 (75.0%)	28 (52.8%)	
Not performed	1 (2.2%)	0 (0.0%)	1 (1.9%)	
**Positive MRI findings**				
Generalized brain atrophy	10 (45.5%)	0 (0.0%)	10 (35.7%)	0.062
Ventriculomegaly	5 (22.7%)	1 (16.7%)	6 (21.4%)	>0.99
Abnormal signal intensity	6 (27.3%)	0 (0.0%)	6 (21.4%)	0.289
Thinning/dysgenesis of corpus callosum	6 (27.3%)	0 (0.0%)	6 (21.4%)	0.289
Cerebellar atrophy	2 (9.1%)	1 (16.7%)	3 (10.7%)	0.530
Lack of/delayed myelination	1 (4.5%)	1 (16.7%)	2 (7.1%)	0.389
Hematoma/hemorrhage	2 (9.1%)	0 (0.0%)	2 (7.1%)	>0.99
Others	8 (36.4%)	4 (66.7%)	12 (42.9%)	0.354
Calcification	1 (4.5%)	0 (0.0%)	1 (3.6%)	--
Cerebral and cerebellar edema	1 (4.5%)	0 (0.0%)	1 (3.6%)	--
Chiari II malformation	0 (0.0%)	1 (16.7%)	1 (3.6%)	--
FLAIR vascular hyperintensity	0 (0.0%)	1 (16.7%)	1 (3.6%)	--
Gliosis	0 (0.0%)	1 (16.7%)	1 (3.6%)	--
Gray matter heterotropia	0 (0.0%)	1 (16.7%)	1 (3.6%)	--
Hemosiderosis	1 (4.5%)	0 (0.0%)	1 (3.6%)	--
Hydrocephalus	1 (4.5%)	0 (0.0%)	1 (3.6%)	--
Ischemia	1 (4.5%)	0 (0.0%)	1 (3.6%)	--
Mesial temporal sclerosis	1 (4.5%)	0 (0.0%)	1 (3.6%)	--
Simplified gyral pattern	0 (0.0%)	1 (16.7%)	1 (3.6%)	--
Subdural hygroma	1 (4.5%)	0 (0.0%)	1 (3.6%)	--
White matter atrophy	1 (4.5%)	0 (0.0%)	1 (3.6%)	--

**Table 4 children-10-01334-t004:** Antiseizure medications (ASMs) and outcome among pediatric patients with genetic testing (N = 53).

	Mutation (N = 45)	No Mutation (N = 8)	Total (N = 53)	*p*-Value
**Current ASMs**				
Levetiracetam	22 (48.9%)	5 (62.5%)	27 (50.9%)	0.704
Topiramate	13 (28.9%)	0 (0.0%)	13 (24.5%)	0.176
Valproic acid	9 (20.0%)	2 (25.0%)	11 (20.8%)	0.665
Phenobarbitone	8 (17.8%)	1 (12.5%)	9 (17.0%)	>0.99
Carbamazepine	7 (15.6%)	1 (12.5%)	8 (15.1%)	>0.99
Lamotrigine	7 (15.6%)	0 (0.0%)	7 (13.2%)	0.577
Clonazepam	6 (13.3%)	0 (0.0%)	6 (11.3%)	0.574
Lorazepam/clobazam	3 (6.7%)	0 (0.0%)	3 (5.7%)	>0.99
Vigabatrin	2 (4.4%)	0 (0.0%)	2 (3.8%)	>0.99
Phenytoin	1 (2.2%)	0 (0.0%)	1 (1.9%)	>0.99
Oxcarbazepine	1 (2.2%)	0 (0.0%)	1 (1.9%)	>0.99
Others	8 (17.8%)	0 (0.0%)	8 (15.1%)	0.333
**Tried ASMs**				
Phenobarbitone	9 (20.0%)	1 (12.5%)	10 (18.9%)	>0.99
Carbamazepine	6 (13.3%)	1 (12.5%)	7 (13.2%)	>0.99
Levetiracetam	5 (11.1%)	1 (12.5%)	6 (11.3%)	>0.99
Topiramate	5 (11.1%)	1 (12.5%)	6 (11.3%)	>0.99
Valproic acid	5 (11.1%)	0 (0.0%)	5 (9.4%)	>0.99
Clonazepam	3 (6.7%)	0 (0.0%)	3 (5.7%)	>0.99
Vigabatrin	3 (6.7%)	0 (0.0%)	3 (5.7%)	>0.99
ACTH	3 (6.7%)	0 (0.0%)	3 (5.7%)	>0.99
Lamotrigine	2 (4.4%)	0 (0.0%)	2 (3.8%)	>0.99
Phenytoin	1 (2.2%)	0 (0.0%)	1 (1.9%)	>0.99
Others	2 (4.4%)	1 (12.5%)	3 (5.7%)	0.394
**Seizure outcome**				
Controlled with ASMs	5 (11.9%)	2 (25.0%)	7 (14.0%)	0.717
Controlled without ASMs	20 (47.6%)	5 (62.5%)	25 (50.0%)	
Partially controlled	7 (16.7%)	0 (0.0%)	7 (14.0%)	
Not controlled	7 (16.7%)	1 (12.5%)	8 (16.0%)	
Death	3 (7.1%)	0 (0.0%)	3 (6.0%)	

**Table 5 children-10-01334-t005:** Genotype–phenotype analysis of 15 patients with frequent mutations * among pediatric patients with genetic epilepsy in the current study.

Mutation	N (%)	Age at Onset	Presentation at Initial Diagnosis	DD	ID	S/LD	PC	FH	Control with ASMs
SCN1A	4 (8.9%)	2 to 12 months	Dravet syndrome (1/4) Febrile then non-febrile GTC (1/4) GTC (1/4) Focal seizures (1/4)	1/4	2/4	3/4	2/4	2/4	2/4
DENND5A	3 (6.7%)	2 to 4 months	Focal seizures (2/3) Apnea and starring (1/3)	3/3	2/3	3/3	2/3	2/3	2/3
KCNQ2	2 (4.4%)	2 days to 4 months	Benign familial neonatal convulsions (1/2) Focal seizures (2/2)	0/2	0/2	0/2	0/2	2/2	2/2
ACY1	2 (4.4%)	4 months	Infantile spasms (2/2)	1/2	2/2	2/2	2/2	2/2	2/2
SCN2A	2 (4.4%)	8 days to 3 months	Blinking of eyes and twitching of the mouth (1/2) Apnea with cyanosis and up rolling of eyes with tonic movement (1/2) Burst-suppression pattern in the EEG (1/2)	1/2	0/2	1/2	0/2	0/2	1/2
PCDH19	2 (4.4%)	9 months to 9 years	Focal seizures (N = 1) Drop-like attacks (N = 1)	2/2	2/2	1/2	0/2	0/2	1/2

* Other non-repeated mutations (one patient each, 2.2%) in the current study; ADAT3, CC2D2A, CHAF1B, DIAPH1, DMBX1, ERCC6, FARS2, GEM 1N4, GNPAT, ITPA, KCNH1, KCNT1, KMT2A, LZTR1, NECAP1, NHLRC1, PACS2, PLA2G6, PNKP, SAMHD1, SLC13A5, SLC2A1, SLC6A8, KCNJ10, ST7, SUOX, SYN1, SYNGAP1, SZT2, THOC2, TSC2, and UFC1. Abbreviations: DD, developmental delay; ID, intellectual disabilities; S/LD, speech/language delay; PC, parents’ consanguinity; FH, family history of epilepsy; GEFS+, genetic epilepsy with febrile seizures plus; GTC, generalized tonic–clonic seizures; ASMs, antiseizure medications; MRI, magnetic resonance imaging.

## Data Availability

Data are available upon request.

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
