# Peer review of "Genotype–Phenotype Analysis of Children with Epilepsy Referred for Whole-Exome Sequencing at a Tertiary Care University Hospital"

_children, 2023, doi:10.3390/children10081334_

Round 1
Reviewer 1 Report
The study is interesting, but there are things that need to be improved:
- the bibliography included in the introduction can be expanded, there are several studies investigating the consequences of epilepsy: for example a 2021 study "Social Cognition in Neurodevelopmental Disorders and Epilepsy" showed that Children and adolescents with focal epilepsy may present a deficit of varying extent in emotion recognition and ToM, compared with TD peers.
- the materials and methods part must be presented better: a "participants" paragraph must be written where the group of patients analyzed must be described (e.g. number of participants, average age), it must be precisely described which statistical tests were used for the variables; the image of the structured questionnaire used should be inserted
- the bibliography in the discussion section can be expanded: there are studies evaluating genetic variants in epilepsy such as "Association between SCN1A gene polymorphisms and drug resistant epilepsy in pediatric patients" and "Long-term outcome of epilepsy in patients with prader–willi syndrome"
Author Response
Response to Reviewer 1:
I would like first to thank the Reviewer for the effort and the pertinently raised questions. Below are the answers to the points raised:
1) The bibliography included in the introduction can be expanded, there are several studies investigating the consequences of epilepsy: for example, a 2021 study "Social Cognition in Neurodevelopmental Disorders and Epilepsy" showed that Children and adolescents with focal epilepsy may present a deficit of varying extent in emotion recognition and ToM, compared with TD peers.
Response: Edited in the revised manuscript and the reference was cited.
2) The materials and methods part must be presented better: a "participants" paragraph must be written where the group of patients analyzed must be described (e.g. number of participants, average age), it must be precisely described which statistical tests were used for the variables; the image of the structured questionnaire used should be inserted
Response: Materials and methods were expanded, and statistical analysis was described and highlighted in the revised manuscript. The image of the original structured questionnaire is inserted (supplementary).
3) The bibliography in the discussion section can be expanded: there are studies evaluating genetic variants in epilepsy such as "Association between SCN1A gene polymorphisms and drug resistant epilepsy in pediatric patients" and "Long-term outcome of epilepsy in patients with prader–willi syndrome"
Response: The association between SCN1A gene polymorphisms and drug-resistant epilepsy was edited in the discussion paragraph of the revised manuscript and the reference was cited. We don’t have patients with PWS, as it can be negative in WES testing, so we think it is not relevant.
Thank you. Sincerely, Fahad A. Bashiri

Reviewer 2 Report
There were some points in the paper that I thought might be clarified with advantage to the general reader. These are:
· Were all your 294 patients first presentations as your clinics, or were existing patients also studied?
· Was any particular selection policy followed in studying only some 17% of your children with epilepsy? The high percentage of those chosen who had abnormalities suggests that they may have been selected on a clinical diagnosis of syndrome basis, particularly when some appear to have had untreated and controlled epilepsies.
· The mean age of your patients was around nine years but the epilepsies had begun in the majority of your patients within the first year of life, and a few of your current patients appear to have died during the course of your study. Is it possible that early death after diagnosis may have prevented you from identifying other more lethal genetic abnormalities?
· In line 150 of your text you use the word ‘insignificant’ in relation to abnormalities. Do you mean that an abnormality is insignificant, or is it not statistically significant at a P < 0.05 level?
· In Table 5 there are, in the top row, a number of unexplained abbreviations, e.g. DD, ID, PC and so on. Also in the Table the dot points symbols seem to wander in and out of vertical alignment.
Author Response
Response to Reviewer 2
I would like first to thank the Reviewer for the effort and the pertinently raised questions. Below are the answers to the points raised:
- Were all your 294 patients first presentations as your clinics, or were existing patients also studied?
Response: We reviewed all the patients following in the clinic who had WES testing.
- Was any particular selection policy followed in studying only some 17% of your children with epilepsy? The high percentage of those chosen who had abnormalities suggests that they may have been selected on a clinical diagnosis of syndrome basis, particularly when some appear to have had untreated and controlled epilepsies.
Response: We agree with this comment. The selection of the patients depends on their clinical presentation especially patients with developmental delay and comorbidities like Intellectual disability. Not all patients with epilepsy in our clinic will be tested routinely with WES.
- The mean age of your patients was around nine years but the epilepsies had begun in the majority of your patients within the first year of life, and a few of your current patients appear to have died during the course of your study. Is it possible that early death after diagnosis may have prevented you from identifying other more lethal genetic abnormalities?
Response: We agree with this comment. However, only 3 patients died out of 53.
- In line 150 of your text you use the word ‘insignificant’ in relation to abnormalities. Do you mean that an abnormality is insignificant, or is it not statistically significant at a P < 0.05 level?
Response: It was not statistically significant ( edited in the revised manuscript)
- In Table 5 there are, in the top row, a number of unexplained abbreviations, e.g. DD, ID, PC and so on. Also in the Table, the dot points symbols seem to wander in and out of vertical alignment.
Response: The abbreviations were explained in the table ( Abbreviations: DD, developmental delay; ID, intellectual disabilities; S/LD, speech/language delay; PC, parents' consanguinity; FH, family history of epilepsy; GEFS+, genetic epilepsy with febrile seizures plus; GTC, generalized tonic-clonic seizures; ASMs, antiseizure medications; MRI, magnetic resonance imaging). Also, The tables were adjusted in the revised manuscript.
Thank you. Sincerely, Fahad A. Bashiri
